# Pesticides: Behavior in Agricultural Soil and Plants

**DOI:** 10.3390/molecules26175370

**Published:** 2021-09-03

**Authors:** Lydia Bondareva, Nataliia Fedorova

**Affiliations:** Federal Scientific Center of Hygiene Named after F.F. Erisman, Federal Service for Surveillance on Consumer Rights Protection and Human Wellbeing, 141014 Mytischi, Moscow Region, Russia; fedorovane@fferisman.ru

**Keywords:** pesticides, agricultural food products, distribution multi-residual method

## Abstract

This review considers potential approaches to solve an important problem concerning the impact of applied pesticides of various classes on living organisms, mainly agricultural crops used as food. We used the method of multi-residual determination of several pesticides in agricultural food products with its practical application for estimating pesticides in real products and in model experiments. The distribution of the pesticide between the components of the soil-plant system was studied with a pesticide of the sulfonylureas class, i.e., rimsulfuron. Autoradiography showed that rimsulfuron inhibits the development of plants considered as weeds. Cereals are less susceptible to the effects of pesticides such as acetamiprid, flumetsulam and florasulam, while the development of legume shoots was inhibited with subsequent plant death.

## 1. Introduction

On the global scale, the damage of agricultural crops is caused by approximately 50,000 species of plant pathogens, 9000 species of insects and mites and 8000 species of pest plants [1,2,3]. This damage to crops in the form of crop loss includes an estimated −13% due to plant pathogens, −14% due to pest insects and −13% due to pest plants [4]. Furthermore, pesticides are indispensable for growing plants, especially for growing economically important crops. According to predictive research, pesticides protect about a third of the world agricultural production. [5]. As is shown in the recent data, about 2 million tons of pesticides are used, including herbicides −47.5%, insecticides −29.5%, fungicides −17.5% and other pesticides −5.5% [6,7].

Pesticides are chemical substances intended for fighting insects, pests, fungi, rodents and microbes. A lot of pesticides are found to be harmful to the health of humans and animals or dangerous for the environment.

The Food and Agriculture Organization of the United Nations (FAO) gives the following definition of pesticides: “Pesticide means any substance or mixture of substances or biological ingredients intended for repelling, destroying or controlling any pest or regulating plant growth” [8,9].

The term pesticide implies more than just a Plant Protection Product (PPP) [10]. Plant Protection Products are “pesticides”, which protect crops or desirable and useful plants. They contain at least one active substance and have one of the following functions:To protect plants or plant products from pests/diseases before or after the harvest (e.g., fungicides, insecticides, molluscicides, nematocides, rodenticides);To influence processes of plant life (e.g., substances affecting their growth, with the exception of nutrients);To preserve plant products (e.g., fumigants);To destroy undesirable plants and their parts or to prevent their growth (e.g., defoliants);To prevent undesirable growth of plants (e.g., herbicides).

The problem of food contamination and food ingredients by residual amounts of pesticides has been urgent for several decades. At present, most agricultural products are being produced according to technologies which rely on the wide application of pesticides. In spite of the efforts of scientists and farmers, “ecological agriculture” cannot provide a necessary amount of food for the world population. 

Taking into account the fact that about 80–85% of residual amounts of pesticides (RAP) enter the human organism with food, special attention is paid to this branch of industry aimed at providing high-quality food for the population [1,2,11].

Since not all countries in the world can supply their population with sufficient agricultural raw materials, as well as with the products of their processing, of special significance are the issues concerning the quality of imported food, including imports from developing countries.

To minimize the risk to human health caused by the residual amounts of pesticides in agricultural products, it is necessary to have vast and reliable information about the level of pollution which could allow one to develop measures to guarantee food safety for the population.

There is a large list of pesticides used in the exporting country; this list also includes combinations of two, three or more active substances in various compositions. Various chemical substances for plant protection are applied several times during the crop growing period. All these above-mentioned factors—as well as the geographical location of the main importing countries, soil and climatic conditions (high temperature, humidity, precipitation, intensive solar radiation), which greatly correlate with the pesticide detoxification rate—influence the RAP level in plant products [11,12,13]. 

The regulations and science of pesticide residues permeate every facet of the food industry, from the farm to the consumer. Comprehensive understanding of the regulatory climate, both domestically and internationally, ensures that the proper precautions are taken to mitigate regulatory and safety risks associated with pesticide use. Furthermore, evaluating pesticide testing screens and capabilities offered by laboratories encourages meaningful results that represent ingredients used for food products at home and abroad. [14].

The implementation of enforcement measures on providing the hygiene safety of pesticides in the consumer market is closely connected with the creation and validation of identification methods and quantitative estimation of their residual amounts. 

The aim of the present review is to study the impact of pesticides on the system of soil and agricultural plants in order to reveal possible negative factors for the resilience of plants used as food for the population.

## 2. Multi-Residual Methods to Determine Pesticides in Agricultural Products

Taking into account the fact that in agriculture, a big list of chemicals is used for plant protection from insects, pests and fungi diseases, as well as for growth stimulation, the control of the composition of these chemicals is a task of major importance. Moreover, if there is a possibility to determine the chemicals used in one sample, this can be a reference in terms of the composition control with the subsequent estimation of their individual and combined impact on living organisms. 

Various sample preparation procedures have been suggested due to a great variety of the pesticides used and inherent matrix complexity. Here, a rather high content of lipids in certain grain samples is said to negatively interfere with the analysis [11,12,13,15].

In recent years, the QuEChERS method (quick, easy, cheap, reliable and safe) has been developed which deserves attention for allowing the determination of residual amounts of pesticides in various matrices. The QuEChERS method was suggested in 2003 [16] for estimating residual amounts of pesticides in foods which do not contain fat, particularly fruit and vegetables. The method based on the initial extraction with acetonitrile and a subsequent stage of purification using the dispersive solid-state extraction (DSPE) decreases the volume of samples and solvents as compared to the traditional techniques based on the redistribution with a liquid. This method is simple, fast and less expensive.

Forty active substances in pesticides were selected for the research (including a number of metabolites), with the substances belonging to different chemical groups (neonicotinoids, tryasols, imidazoles, pyrethroids, organophosphorus compounds, strobilurins, etc.) [17,18].

The identification was performed taking into account the retention time, presence of characteristic ions in the mass spectra (GC-MS method) and product ions (LC-MS/MS) and the area ratio of the chromatographic peaks, which are related to the characteristic ions. 

To choose the conditions for detection (the LC-MS/MS method), optimization was implemented for 23 active substances (Figure 1), i.e., scanning in the positive and negative ionization mode using a source of electrostatic spraying. For each analyte, several product ions were obtained, formed after the destruction of the parent ion; the compounds were identified by the multiple reaction monitoring (mass-transfer). 

The method of gas chromatography-mass spectrometry was used to estimate the residual amounts of 18 compounds (Figure 2). In the first research stage, in order to identify the whole list of substances, with all of them being present, use was made of the full scanning mode (by full ion current) in the range from 50 to 500 atomic mass units with the automatic library search “NIST”.

The application of the sample preparation method for QuEChERS excluded a number of active substances from the analysis; these substances, according to their structure, physical and chemical properties and ability for metabolic degradation with the formation of numerous metabolites, cannot be analyzed by the given technique.

It is worth noting that the original method QuEChERS [15,16] is only slightly different from the main procedure being applied at present to matrices with the low (2–20%) or high (>20%) fat content. The main difference is in the purification stage since even a small amount of fat is also co-extracted from the matrix and can affect the subsequent chromatographic analysis [17,18].

The method for preparing QuEChERS samples is very widely developed both for analyzing new media (of different origin, composition, physical and chemical properties) and for studying numerous combinations of pesticides, followed by GC-MS and LC-MS/MS detection [19,20,21]. 

For example, the developed methods of multi-residual determination of pesticides were also tested in estimating the contamination of grain crops by residual amounts of pesticides. The attention was focused on wheat, maize and rice, the grain crops which amount to 88% of the world grain production [22]; these methods were also successfully applied to determine pesticides in a number of tropical and dried fruit.

## 3. Herbicide-Absorption and Translocation in the Soil-Plants Systems 

Pesticides can be characterized by various degrees of toxicity for target and non-target organisms [23,24,25,26,27,28,29,30,31,32,33]. Due to their cumulative properties, many pesticides [32,33,34,35] circulate in ecosystems, can be accumulated by many living organisms and even migrate through them along food chains. For recognizing the impact of a pesticide, certain biological specimens, individuals, species and communities are predominantly used as models for estimating harmful effects. Pesticides can penetrate into an organism (1) depending on the species and peculiarities of metabolism, and also, (2) depending on the level of susceptibility to toxins [36,37]. However, if a chemical substance has already penetrated into an organism, it must be able to fight it in order to neutralize or minimize its harmful impact by means of biotransformation, conjugation, isolation and/or release into the environment or by means of a combination of the above mechanisms. All these efforts are directed towards preventing or minimizing the harm for the organism [38,39]. 

Many agricultural areas with humic-sandy and loamy-sandy soils are also used for the extraction of water for drinking water supply. The pesticide concentrations per depth in soil are highly variable due to local differences in transport, adsorption and transformation. Measurements both in the subsoil and in the upper groundwater are scarce, also due to sampling problems. The methods of pesticide analysis in soil samples are often not sensitive enough to measure micro concentrations of the substances [40].

A literature review of tests of four simulation models for pesticides in the soil-plant system was presented by Van den Bosch and Boesten [41]. The frequently occurring shortcomings in the experimental data sets used for the tests are the following:Poor characterization of soil profiles and compositions of the layers.Weather conditions not quantified (rainfall/irrigation pattern).No information on the crop growth during the experiment (soil cover/leaf area index, root development).Few soil-sampling field experiments, low number of samples and too shallow soil sampling (only the top layer).Too short duration of the experiments to study deeper movement.Too low sensitivity of the concentration measurements in soil (high determination limit).No site-specific measurements on adsorption and rate of transformation in soil as model inputs (e.g., data taken from handbooks and databases).Pesticide uptake by the plant roots set to zero (because of the lack of data) [42,43].

The research was carried out directly with rimsulfuron and with a labeled ^14^C preparation [44].

Two species of plants were used in the experiments: two-week old shoots of *Sinapis**arvensis* L. as weed plants, and two-week old seedlings of *Zeamays* L., 1753 as crop plants.

Soil was taken from the secondary forest of the Moscow Region. It was classified as sod-podzolic with sandy-clay texture. After the litter removal, the soil was collected from the upper 20 cm layer, dried at room temperature and sieved through a 2 mm mesh. The soil was fertilized with 30 g NPK 4-14-8 in each microcosm and limed to pH 5 to 5.5.

Table 1 presents the parameters of the plants before and after the introduction of the herbicide.

Table 2 presents the estimation data on the proportion of the radioactivity distribution over the components of the model systems.

The presented results show that a larger portion of radioactivity is found in the soil both in the systems with and without plants (from 55.0% to 99.5%). The root part of the soil, rhizosphere, was analyzed separately. The amount of radioactivity in this part is approximately the same for the systems with plants, including weeds, given the degradation of the above-ground part (from 20.5% to 22.5%). 

It is interesting to know whether the calculated concentration profiles of the substances are sensitive to the way of the soil layer simulation.

Figure 3a,b shows the distribution results for rimsulfuron and ^14^C over the studied layers of the system.

To analyze the soil in the model experiments, we used the method of multi-residual determination of pesticides, described above. Matrix effects (MEs) are one of the main aspects that must be addressed when evaluating a multi-residue method for pesticide analysis. The procedure also was based on the quick, easy, cheap, effective, rugged and safe (QuEChERS) sample preparation method. The choice of the buffer, type of extract solvent, shaking time and dispersive solid-phase extraction (d-SPE) clean-up were optimized. The study showed that the content of pesticides analyzed by our method was lower than the detection limit, with the exception of rimsulfuron, which we had introduced.

The results show the distribution of the pesticide and radiocarbon 14 days after the application of the labeled preparation to the experimental system enclosed in a microcosm.

The highest pesticide and radiocarbon content was found in the uppermost soil layer, ~54% of the amount applied. With the distance from the soil surface, the labelled pesticide content apparently decreased. The resulting distribution was not unusual due to the main downward transport of fluid flows between the soil particles.

Insignificant amounts of rimsulfuron and radiocarbon were also detected in drainage waters, which were collected in a tray installed beneath a cylinder in which the test soil was placed: ~0.5% of ^14^C and rimsulfuron.

The half-life of rimsulfuron was determined as being 22 days in laboratory conditions. The linear adsorption coefficient was K_d_ = 18 mg·L^−1^. The calculated value of Gibbs free energy was <50 kJ·mol^−1^, which indicates mainly physical preparation adsorption with soil particles under exothermic conditions. The results have been statistically processed [44]. 

To assess the contribution of the pesticide loss as a result of metabolism, studies were conducted to identify the degradation rate for the initial pesticide [45,46,47,48,49,50]. 

A research team in [51] studied adsorption, degradation and leaching migration characteristics of chlorothalonil in different soils. The results show that the adsorption of chlorothalonil in clay and sandy soils can be characterized by the Freundlich equation. The adsorption coefficient (K) was found to be 6.7158 and 1.2568, respectively. The residual degradation kinetics of chlorothalonil in both soils corresponds to the first-order kinetics degradation equation. As the concentration of chlorothalonil increased, the higher the residual amount of chlorothalonil in the soil, the slower was the degradation rate and the longer the half-life. In the soil column, chlorothalonil could not easily move and migrate in the two soil columns. The highest residual residues were in the range from 0 to 10 cm (the topmost), with the following decrease. The correlation analysis showed that the adsorption and leaching of chlorothalonil in the two soils may be affected by a combination of factors such as soil organic matter content, clay content, cation exchange capacity and soil pH value. This leads to a great risk of the groundwater contamination, and thus, it should be paid serious attention [51].

The adsorption, desorption and leaching potential of glyphosate and aminomethylphosphonic acid [52], adsorption of dieldrin by parent and processed montmorillonite clays [53] and other pesticides were studied including the biodegradation of pesticides by soil bacteria [54,55,56].

The content of ^14^C in the leaves varied from 9.3 to 11.4, with this amount being higher in the weed leaves. With the simultaneous presence of two plant species, the amount of ^14^C in the leaves was comparable with the amount of the radioisotope in the leaves of *Sinapis arvensis L*., despite the fact that there was intense necrosis of the entire leaf surface. In this case, the radioactive isotope appeared to be incorporated into the leaf structure, intensely affecting the cells. 

The physiology of the maize leaves did not undergo any noticeable changes, and the detected content of ^14^C was likely to be present only on the surface, or in the uppermost layer of the epidermis of the leaf blade.

The stem in the studied systems can be assigned solely the role of a conductor of the preparation between the leaves treated at the very beginning of the experiment and the soil in which the plants grew.

The herbicides applied to the plants can undergo the following [47,48,49,50,51]: (1) volatilize off the leaf surface before absorption, (2) be washed off the leaf surface before absorption, (3) photodegrade before absorption, (4) remain on the leaf surface, without being absorbed, (5) penetrate the cuticle, and remain tied up in the cuticle, (6) penetrate the cuticle, and enter the apoplast or symplast and (7) be subject to translocation and metabolism. This may be caused by intensive photosynthesis and transpiration processes in plant edible parts and by their growing conditions: located close to the soil surface, they are attacked by insects more frequently, and hence, larger amounts of pesticides are applied to protect them.

Whole-body autoradiography is widely used to trace the routes of molecules in metabolism. First, a radioactive tracer (^14^C-rimsulfuron) is administered to an organism by ingestion or injection. After a period of time, individual samples of tissue are removed and pressed directly against an X-ray film for several days, to expose the film wherever the radioactivity has become concentrated. 

The film is then developed and examined, mostly using a microscope. This process is used to trace the uptake of nutrients by the plants leaves or buds from the soil. 

Figure 4 presents the autoradiography results for the maize leaves: the left panel shows transverse cross-section, while the right panel is the view from the top. 

As can be seen from the presented results, a larger amount of radiocarbon is found in the leaf part which is closer to the stem (the lighter part of the image), with some inclusions in the rest of the leaf. When examining the image obtained for the leaf cross section using a scanning microscope (Figure 4A,B), the part of the leaf facing the surface was found to contain more ^14^C than the inner part of the leaf.

This is likely to be due to the fact that rimsulfuron, when sprayed, first covers the outer parts of the plant leaves. This is where the primary effect of the preparation is exerted on the physiological functions of the plants, including the impact associated with the absorption and transformation of the introduced herbicide.

Figure 5 presents the accumulation of ^14^C by the intracellular space of the maize leaf.

Figure 6 shows the autoradiography results for the weed plant *Sinapis arvensis* L. in the system with the ^14^C-labelled rimsulfuron treatment, the exposure time being 32 h. 

The accumulation of radiocarbon in the plant tissues is dark colored in Figure 5 and Figure 6. One can see in Figure 5 that ^14^C is present in almost all the organelles of the intracellular space of the maize leaf *Zea mays* L., 1753. The image presented in Figure 6 shows that radiocarbon is detected in the root, stem and remaining leaf of the shoot of the *Sinapis arvensis* L. After additional 12 h following the treatment, all the shoots of *Sinapis arvensis* L. died and shed onto the soil surface, to be removed for the purity of the experiment.

The question arises as to what is the origin of these processes?

When considering the toxic effects of pesticides on living organisms, it is necessary to take into account the stability of the preparations in the environment when exposed to humidity, UV radiation, changes in temperature, etc., since the more stable the pesticide, the greater the level of its accumulation in an organism. Pesticides such as chlordane, dieldrin, hexachlorobenzene thiobencarb and endrin are reported to be resistant to degradation (persistent organic pollutants) and they remain in the environment for a long time. In addition, persistent pesticide residues can accumulate in the organism and reach the bioconcentration more than 70,000 times higher than the initial concentrations [56,57,58,59,60,61,62,63]. 

Earlier, we conducted research on samples of honey and other honey products. Studies were carried out to reveal the presence of residues of a number of pesticides, including neonicotinoids. The possible concentration of residues depends on the amount of the pesticide used (excessive or moderate use), which reflects the accumulation and toxic effects of such residues on pollinators (bees) and other insects [64,65,66,67]. 

The main way is the mechanistic route of the pesticide uptake, starting from the moment of the pesticide application, followed by photodegradation, and absorption by plant parts (stem, leaves or fruit) or sorption at the soil level. This means that pesticides enter the soil, where they undergo biodegradation, chemical decomposition (pH, humidity and temperature) and biodegradation (enzymes of bombardment).

The pesticide residues and decomposition by-products penetrate through the roots into the entire plant parts, causing some detrimental effects on soil and plants. These effects include overproduction of ROS, oxidative stress, DNA damage, photosynthetic blockade, necrosis, chlorosis, leaf curl and ultimately, plant death. An example of this process is the research described in the section above.

Generalized hazardous effects of pesticides and the most common toxic effects of the main types of pesticides (insecticides, herbicides and fungicides) on the soil and plants are listed in Table 3 [68].

During the study aimed at assessing the toxicity, with some agricultural crops from the bean and cereal families subjected to a number of pesticides, ambiguous results were obtained, which are presented in Table 4.

When bean and cereals were treated with pesticides, more stable morphological parameters were observed in experiments with Secále and Avéna (Table 4). The development of bean shoots ceased; the plants began to wither, and their development almost stopped (Figure 7). In the case of cereals, the shoot development visually deteriorated (Figure 8), mainly due to some dried leaf tips.

To assess the efficiency of the photosynthetic apparatus of plants and its resistance to various external influences, methods were developed for considering the intensity of the delayed fluorescence (DF) [69,70,71,72,73,74]. 

DF of photosynthetic organisms, discovered in 1951 by B. Strehler and W. Arnold (USA), is now widely used to study the mechanisms of photosynthetic reactions and their relationship with the physiological state of plants [70].

DF is a biophysical method which provides information about the functioning of the primary reactions of photosynthesis in intact objects or the afterglow of photosynthetic organisms.

Briefly, the origin and mechanism of the DF in photosynthetic organisms can be represented as follows.

A comparative study of the emission spectra of fast (10^−9^ s) and delayed (in ms) fluorescence showed their similarity. On this basis, it was concluded that DF occurred during the radiative deactivation of the first singlet excited state of chlorophyll [70,73,74].

At the same time, in contrast to the rapid fluorescence of plant chlorophyll, which decays in a time of the order of 10^−8^–10^−9^ s, the duration of the DF significantly exceeds the intrinsic time of the singlet excited state of chlorophyll. This shows that DF is due to the secondary excitation of chlorophyll during reverse reactions formed in the light of photoproducts [69,73,74].

The kinetics of the DF are very complex and multicomponent. This is due to the fact that the stages of stabilization of the stored light energy are reversible and can generate an excited state of chlorophyll [71].

It has been established that DF of green plants occurs mainly in the reaction centers of photosystem II and a very weak DF can be observed in the photosystem.

During the recombination of the excited molecules of the reaction center, a part of the resulting energy is transferred to the molecules of light-harvesting chlorophyll, which emits it in the form of quanta of DF.

The DF method is characterized by high sensitivity, reliability, rapidity and ability to automate the information obtained from intact objects in the field [71,74].

Using delayed fluorescence to detect the effect of the applied pesticides on the studied plants of the bean and cereal families, a decrease in the intensity of the DF was observed (Figure 9).

Based on the results of the entire experiment and intensity of the DF, one can conclude that the cereal plants are more resistant to treatment with the studied pesticides than the bean plants. The indices of DF are decreased by a factor of 2 or less for the cereals, while for the beans, a complete suppression of the indices is observed. In the system with rye, the intensity of DF hardly changed when acetamiprid, flumetsulam and florasulam were added to the system.

The above presented studies and discussions highlight that pesticide residues cause direct and indirect damage to fauna, flora, physicochemical and biological properties of agricultural soils. In addition, they can reduce enzymatic activity and suppress microbial communities in the soil. Pesticides can cause chlorosis, leaf curl/necrosis and photosynthetic impairment, including oxidative stress [74,75,76,77]. Different classes of pesticides lead to a suppression of nitrogen metabolism by increasing or decreasing the activity of certain enzymes. In addition, the pigmentation of the leaves may change, and grains may stop developing. The validation of the method for multi-residual determination of pesticides belonging to different classes in terms of the chemical structure and properties, and having different types of effects, makes it possible to simultaneously detect the presence of certain substances in the environments of different types.

## 4. Conclusions

The implementation of enforcement measures on providing the hygiene safety of pesticides to the consumer market is closely connected with the creation and validation of identification methods and quantitative estimation of their residual amounts. 

The aim of the present review was to study the impact of pesticides on the “soil-agricultural plants system” in order to reveal possible negative factors for the resilience of plants used as food for the population.

(1) The method of multi-residual determination of pesticides in agricultural crops was considered. Forty active substances in pesticides were selected for the research (including a number of metabolites), with the substances belonging to different chemical groups (neonicotinoids, tryasols, imidazoles, pyrethroids, organophosphorus compounds, strobilurins, etc.). The identification was performed taking into account the retention time, presence of characteristic ions in the mass spectra (GC-MS method) and product ions (LC-MS/MS) and area ratio of the chromatographic peaks which are related to the characteristic ions. The method QuEChERS was used to prepare the samples of agricultural products. This methodological approach is very promising in the research of pesticides that have different physical and chemical properties and ways of their intake by the organism, as well as various types of behavior in the soil-water-plant system.

(2) In the model experiments to study the migration and translocation in the soil-plant system (the agricultural crop *Zeamays* L., 1753 and the weed *Sinapisarvensis* L.), using rimsulfuron and labelled ^14^C-rimsulfuron, a larger portion of radioactivity was found in the soil both in the systems with the plants and without them (from 55.0% to 99.5%); in all the systems, the radioactive content in the root zone of the soil, i.e., in rhizosphere, was from 20.5% to 22.5%; the content of ^14^C in the leaves varied from 9.3% to 11.4%. The degradation of the weed *Sinapisarvensis* L. followed by its decomposition was revealed by the method of autoradiography. However, in the leaves of maize (*Zeamays* L., 1753), these processes spread both on the leaf surface and in its intracellular compartment, with no visible changes in the vital functions of the plants found.

(3) Based on our results and on the data from the information sources, the adsorption of pesticides in soil was found to follow the regularity described by the Freundlich equation.

(4) During the study of the toxic impact of acetamiprid, flumetsulam and florasulam on some crops of the bean family and cereal family, ambiguous results were obtained: the bean family was affected by the considered pesticides to the highest extent (including the complete inhibition of the plant shoots), while the cereal plants were hardly affected. 

## Figures and Tables

**Figure 1 molecules-26-05370-f001:**
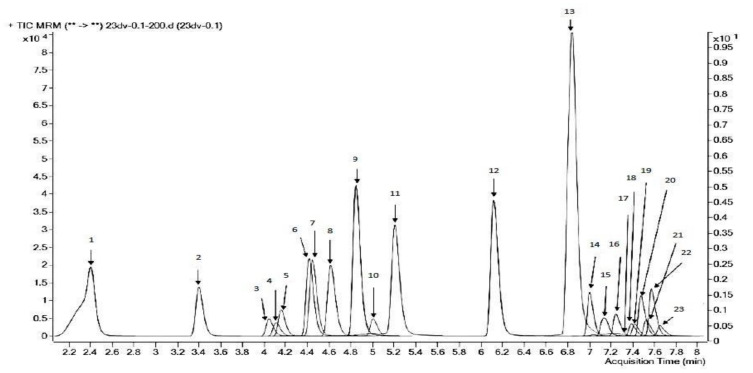
Chromatogram of the model solution of 23 pesticides with the concentration of 0.1 µg·ml^−1^, the electrospray ionization source (ESI) in the positive ion mode (TIC—total ion current): 1—omethoate, 2—thiamethoxam, 3—imidacloprid, 4—clothianidine, 5—flumetsulam, 6—dimethoate, 7—acetamiprid, 8—rimsulfuron, 9—thiacloprid, 10—florasulam, 11—thiabendazole, 12—carboxim, 13—spiroxamine, 14—fluxapiroxade, 15—fluopyram, 16—epoxiconazole, 17—iprodione, 18—kresoxym-methil, 19—penconazole, 20—pyraclostrobin, 21—prochloraz, 22—trifloxystrobin, 23—ipconazole. On the X axis is the time (min), on the Y axis is the peak intensity.

**Figure 2 molecules-26-05370-f002:**
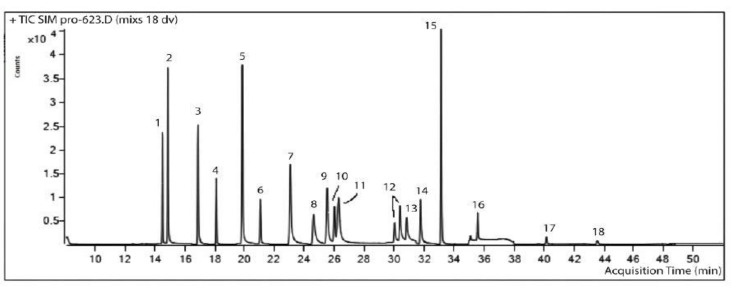
Chromatogram of the model solution of 18 pesticides with the concentration of 0.1 µg·ml^−1^. The Selected Ion Mode (SIM): 1—diazinon, 2—chlorothalonil, 3—mefenoxam, 4—malathion, 5—cyprodinin, 6—fipronil, 7—fludioxanil, 8—flutriafol, 9—prothioconazole-desthio, 10—fipronil sulfone, 11—cyproconazole, 12—propiconazole, 13—tebuconazole, 14—epoxiconazole, 15—bifentrin, 16—lambda-cyhalothrin, 17—alpha-cyperomenthrin, 18—esfenvalerate. On the X axis is the time (min), on the Y axis is the peak intensity.

**Figure 3 molecules-26-05370-f003:**
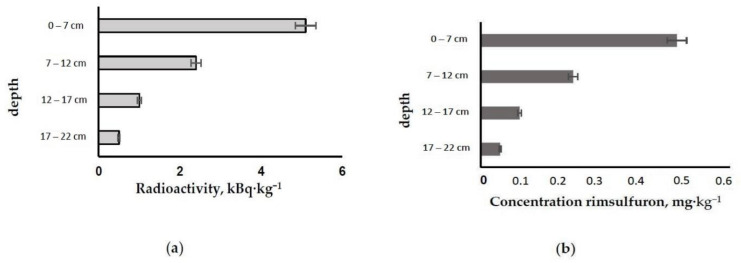
Distribution of ^14^C isotope, kBq·kg^−1^ (**a**) and pesticide (**b**) over the soil profile of the experimental system [44].

**Figure 4 molecules-26-05370-f004:**
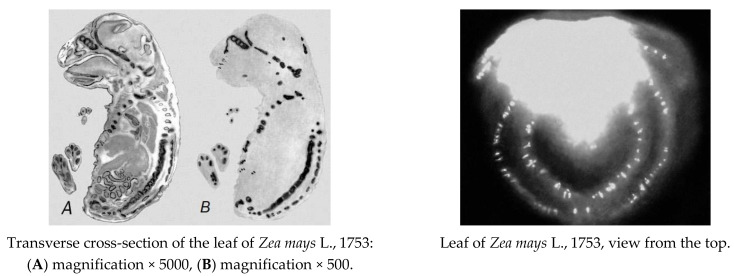
Distribution of the labelled rimsulfuron over the leaf of *Zea mays* L., 1753 [44].

**Figure 5 molecules-26-05370-f005:**
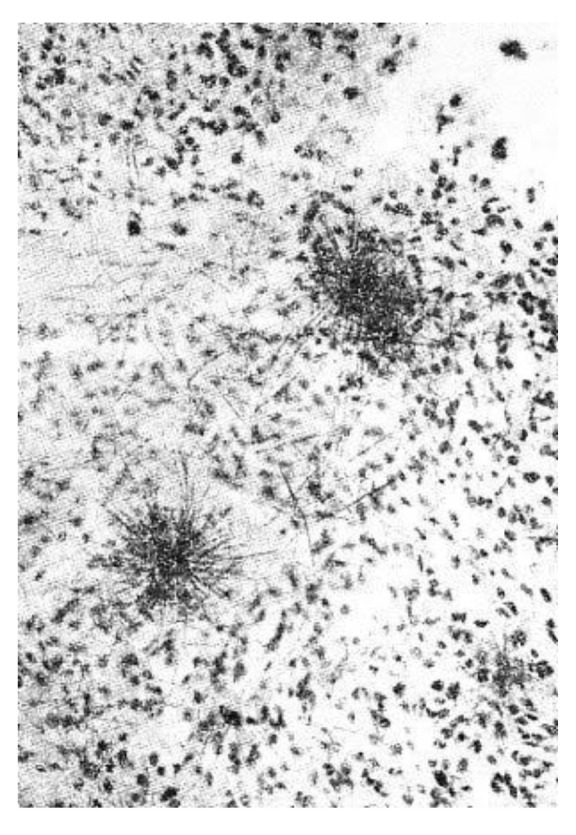
Distribution of ^14^C in the intracellular space of the leaf of *Zea mays* L., 1753 [44].

**Figure 6 molecules-26-05370-f006:**
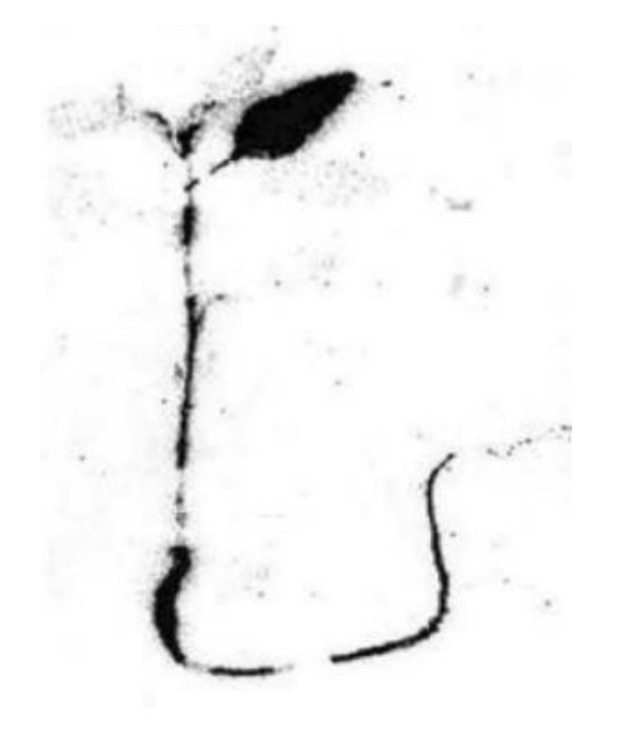
Autoradiography of the shoots *Sinapis arvensis* L., 32 h after the treatment [44].

**Figure 7 molecules-26-05370-f007:**
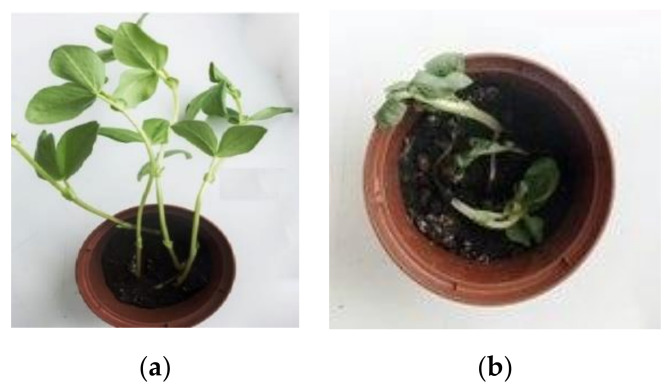
Comparison of the physical appearance of the bean shoots (*Vícia fába*): (**a**) control, (**b**) after treatment with the pesticide (acetamiprid).

**Figure 8 molecules-26-05370-f008:**
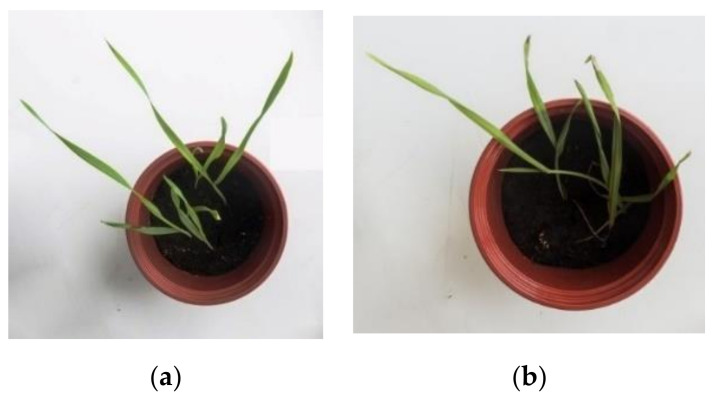
Comparison of the physical appearance of the rye shoots (*Secále*): (**a**) control, (**b**) after treatment with the pesticide (acetamiprid).

**Figure 9 molecules-26-05370-f009:**
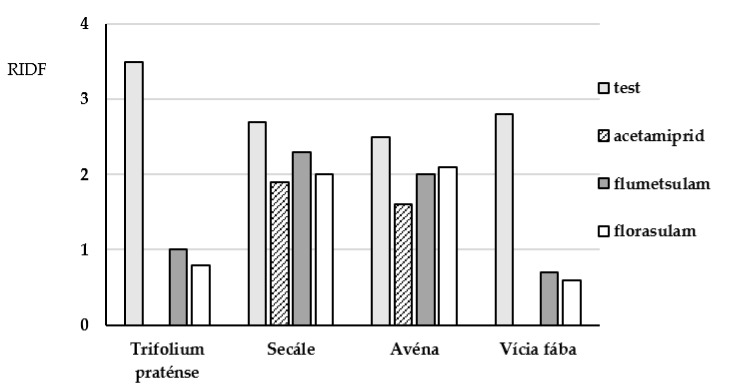
Dependence of the dynamics of changes in the relative index of delayed fluorescence (RIDF) on the pesticide concentration.

**Table 1 molecules-26-05370-t001:** Dynamics of changes in the plant parameters [44].

System	Initial Parameters (*n* = 5)		Parameters at the End of the Experiment (*n* = 5)			
	Length, cm	Weight, g	Length, cm	Weight, g		
				Wet	Dry	Dry/Wet, %
*Sinapis arvensis* L.,	1.5 ± 0.2	0.7 ± 0.4	1.6 ± 0.4	0.6 ± 0.2	0.03 ± 0.01	5.0
*Zea mays* L., 1753	5.4 ± 1.2	4.7 ± 1.7	8.3 ± 2.1	10.5 ± 3.0	3.2 ± 1.4	30.5

*n:* number of measurements.

**Table 2 molecules-26-05370-t002:** Distribution of the radioactivity over the components of the soil-plants systems [44].

System	Content ^14^C, kBq (% from the Introduced One)				
	Soil (the Whole Amount)	Rhizosphere	Stems (the Whole Amount)	Leaves (the Whole Amount)	Tube for ^14^C-CO_2_ Capture
*Sinapis arvensis* L.,	53.8 ± 0.9 (56.6)	21.4 ± 0.6 (22.5)	7.0 ± 1.3 (7.4)	10.8 ± 0.8 (11.4)	2.0 ± 0.7 (2.1)
*Zea mays* L., 1753	57.1 ± 0.7 (60.1)	19.5 ± 0.7 (20.5)	6.1 ± 0.9 (6.4)	8.8 ± 1.1 (9.3)	3.5 ± 1.0 (3.6)
*Sinapis arvensis* L. + *Zea mays* L., 1753	52.6 ± 0.8 (55.4)	20.9 ± 1.1 (22.0)	7.0 ± 1.1 (7.4)	10.5 ± 1.2 (11.1)	4.0 ± 0.9 (4.2)
Soil (sod-podzolic with sandy-clay texture)	94.5 ± 0.7 (99.5)	-	-	-	0.5 ± 0.2 (0.5)

**Table 3 molecules-26-05370-t003:** Toxic effects of pesticides on agricultural soils and plants.

Pesticide Type	Toxic Effects	
	Soil	Plants
Insecticides	Destruction of microbial structural proteins, symbiotic attributes reduction, change in soil chemistry and enzymatic activity	Reduction in grain protein content, blockage of stomatal conductance and alterations in the photosynthetic process
Herbicides	Reduction in the soil nutrient availability and suppression of phosphatase and nitrogenase activities	Alteration of the physiological and biochemical plant efficiency, increasing the susceptibility of plants to diseases
Fungicides	Interruption of phosphatase, urease and dehydrogenase activities and inhibition of the nitrifying bacterial growth	Reduction in chlorophyll and carotenoid concentration, destruction of chloroplasts, stomatal closure and electron transfer suppression

**Table 4 molecules-26-05370-t004:** Changes in the morphological parameters for agricultural crops: rye, oats, bean and clover upon the application of pesticides (*n* = 15, *p* = 0.95).

Pesticide	Plant	Size of the Plant, Length, cm			
		Above-Ground Part L_cp_. ± ∆		Root Part L_cp_. ± ∆	
		Test	0.001 mg·kg^−1^	Test	0.001 mg·kg^−1^
acetamiprid	clover (*Trifolium praténse*)	4.8 ± 0.4	0	1.3 ± 0.3	0
flumetsulam		4.8 ± 0.4	3.3 ± 0.5	1.3 ± 0.3	0.9 ± 0.3
florasulam		4.8 ± 0.4	2.6 ± 0.4	1.3 ± 0.3	0.5 ± 0.3
acetamiprid	rye (*Secále*)	26.0 ± 1.0	20.0 ± 2.0	9.5 ± 0.6	7.3 ± 0.7
flumetsulam		26.0 ± 1.0	15.0 ± 1.0	9.5 ± 0.6	10.0 ± 1.0
florasulam		26.0 ± 1.0	11.3 ± 0.7	9.5 ± 0.6	5.1 ± 0.9
acetamiprid	oats (*Avéna*)	21.0 ± 1.0	20.0 ± 0.8	7.0 ± 0.4	5.5 ± 0.6
flumetsulam		21.0 ± 1.0	18.7 ± 0.7	7.0 ± 0.4	4.8 ± 0.5
florasulam		21.0 ± 1.0	18.1 ± 0.8	7.0 ± 0.4	5.7 ± 0.3
acetamiprid	bean (*Vícia fába*)	3.1 ± 0.6	0	0.7 ± 0.2	0
flumetsulam		3.1 ± 0.6	0	0.7 ± 0.2	0
florasulam		3.1 ± 0.6	0	0.7 ± 0.2	0

## Data Availability

The data presented in this study are not available.

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
