# Peer review of "Pesticides: Behavior in Agricultural Soil and Plants"

_molecules, 2021, doi:10.3390/molecules26175370_

Round 1

Reviewer 1 Report

The authors reviewed the effects of pesticides on food crops and the detection methods of multiple pesticide residues.  However, I have some comments and suggestions and would like to see the authors’ corrections.

Corrections:

  • There are too many sections throughout the article, and the context is weak, so it is suggested to modify and integrate.
  • It is suggested that the full text should be in a unified reference format. If permitted, please change it.
  • Table 2 and 3 are not shown in the text, Please complete them.
  • The sharpness of the pictures in the article is insufficient, for example, Figure 1 and Figure 2, and the ordinate Counts in Figure 2 falls outside the figure. In addition, the formats of the same kind of pictures should be unified as far as possible, for example, the formats a and b in Figure 3 should be adjusted.
  • Please standardize the use of punctuation marks in the whole text, such as Table 4, Please correct if it is a writing error.
  • Please give the meaning of each character in the table, for example, "n" and "Initial Parameters" in table 1 in line 183. If this table is not the result of the author's experimental study, please give the source.
  • Line 63-69, the concepts of importing country and exporting country are reversed.
  • "1" and "2" in lines 138-148 are not supported by the literature.
  • The 3.1 section should not only describe the adsorption and migration characteristics of herbicides, but also describe the adsorption and migration characteristics of other types of pesticides.
  • As an overview, line 292 contains too few examples.
  • Lines 316-320 of the conclusion are not mentioned in the article, Please check it.
  • The author did not indicate the direction of development in this field, please fill in if possible.

Author Response

There are too many sections throughout the article, and the context is weak, so it is suggested to modify and integrate.

It is suggested that the full text should be in a unified reference format. If permitted, please change it.

  • Thank you. We changed: some parts - deleted, some fragments – added.

Tables 2 and 3 are not shown in the text, please complete them. – This is our mistake. We changed.

The sharpness of the pictures in the article is insufficient, for example, in Figure 1 and Figure 2, and the ordinate Counts in Figure 2 falls outside the figure. In addition, the formats of the same kind of pictures should be unified as far as possible, for example, the formats a and b in Figure 3 should be adjusted.

  • Thank you. We inserted a better version of Figures 1 and 2.

Please standardize the use of punctuation marks in the whole text, such as Table 4, Please correct if it is a writing error.

  • Thank you. We changed

Please give the meaning of each character in the table, for example, "n" and "Initial Parameters" in table 1 in line 183. If this table is not the result of the author's experimental study, please give the source.

  • We added

Line 63-69, the concepts of importing country and exporting country are reversed.

  • OK.

"1" and "2" in lines 138-148 are not supported by the literature.

  • Added

The 3.1 section should not only describe the adsorption and migration characteristics of herbicides but also describe the adsorption and migration characteristics of other types of pesticides.

As an overview, line 292 contains too few examples.

Lines 316-320 of the conclusion are not mentioned in the article, Please check it.

  • Thank you. We changed “Conclusion”

The author did not indicate the direction of development in this field, please fill in if possible.

  • Thank you. We made to some changing 

Reviewer 2 Report

The article Pesticides: Behavior in Agricultural Soil and Plants is suitable for publication in Molecules mainly due to the relevance of the problem of pesticides presence in the environment. I believe my detailed comments will be useful.

Also I suggest to use native speaker for English correction.

Abstract

In my opinion you should extend you abstract.

Also the sentence from line Lines 9-10(Special attention is paid to the validation of the multi-residual method for determing 40 items of pesticides in agricultural food products)   should be rewritten.

Introduction

Line 47 should be rewritten into: the problem of food contamination and food ingredients…..

Also in the end of this chapter please give provide justification for such your review. Please give information to whom this article is addressed

Line 111: In the ≪hard≫ mode- this is not clear.

Is it possible to improve resolution of figure 1 and figure 2?

 Lines 138-143 must be rewritten, method I, method II, method III. It is difficult to understand scheme.

Figure 3: depth instead deep probably

Give description of figures 5 and 6

Table 6 You wrote: rye, oat, bean, and clover. I cannot see rye, oat, bean, and clover it in your table.

Description of figure 9 is not sufficient.

Conclusions not conclusion

Chapter Conclusions should be definitely rewritten. It must be extended, just give some specific results.

Author Response

In my opinion you should extend you abstract.

Also the sentence from line Lines 9-10(Special attention is paid to the validation of the multi-residual method for determing 40 items of pesticides in agricultural food products)   should be rewritten.

  • Dear reviewer, Thank you. We agree and changed

Introduction

Line 47 should be rewritten into: the problem of food contamination and food ingredients…..

Also in the end of this chapter please give provide justification for such your review. Please give information to whom this article is addressed

  • Thank you. We changed

Line 111: In the ≪hard≫ mode- this is not clear.

  • Removed

Is it possible to improve resolution of figure 1 and figure 2?

  • Thank you. We inserted a better version of Figures 1 and 2.

 Lines 138-143 must be rewritten, method I, method II, method III. It is difficult to understand scheme.

  • Thank you. We changed

Figure 3: depth instead deep probably

  • We agree and changed

Give description of figures 5 and 6

  • Added

Table 6 You wrote: rye, oat, bean, and clover. I cannot see rye, oat, bean, and clover it in your table.

  • Thank you. We added. This is our mistake.

Description of figure 9 is not sufficient.

  • We added a description of the method with reference.

 Conclusions not conclusion

Chapter Conclusions should be definitely rewritten. It must be extended, just give some specific results.

  • We changed. Thank you

Round 2

Reviewer 1 Report

  • The effects of pesticides on food crops and detection methods of pesticide residues were reviewed. The author has revised the article, but some problems still exist, hope to see the author's modification.

    Corrections:

    • Lines 231-242. Can the problem mentioned here be solved using the method in the text? Please discuss this section briefly in the article.
    • Line 198,Herbicide - Absorption and Translocation” It is suggested that soil systems be mentioned in the title to make it relevant to the content of the article.
    • Line426, please unify the units in Table 4.
    • Please standardize the use of punctuation in the whole paper again, for example, punctuation should be added at the end of line 405 "and biodegradation (enzymes of Bombardment)".
    • Line 427 "pesticides (n=15, p=0,95)." Is the comma used properly here?
    • Line443,"Delayed fluorescence (DF)", please use the abbreviation when it first appears in the passage, if the abbreviation of the word is not used later in the passage, please leave it out.
    • Line 283,Please complete the first column data of "Soil" in Table 2. In addition, please rearrange Table 2 according to the journal requirements.

Author Response

Dear Reviewer. Thank you

Lines 231-242. Can the problem mentioned here be solved using the method in the text? Please discuss this section briefly in the article.

Yes. We added

Line 198, “Herbicide - Absorption and Translocation” It is suggested that soil systems be mentioned in the title to make it relevant to the content of the article.

We agree

Line426, please unify the units in Table 4.

  1.  

Please standardize the use of punctuation in the whole paper again, for example, punctuation should be added at the end of line 405 "and biodegradation (enzymes of Bombardment)".

We made

Line 427 "pesticides (n=15, p=0,95)." Is the comma used properly here?

changed

Line443,"Delayed fluorescence (DF)", please use the abbreviation when it first appears in the passage, if the abbreviation of the word is not used later in the passage, please leave it out.

We changed

Line 283,Please complete the first column data of "Soil" in Table 2. In addition, please rearrange Table 2 according to the journal requirements.

We added type of the soil
